# Approximating Drift-Diffusion Models for User Decisions Under Nudging and External Information

**Gustavo Grivol** [1]    **Hanna Halaburda** [1]    **Alexander Tuzhilin** [1]

## Abstract

Modeling decision-making outside of controlled environments requires accounting for asynchronous, exogenous signals, such as notifications or algorithmic feeds, that dynamically alter user response times. Standard Drift-Diffusion Model (DDM) framework become analytically intractable when drift rates vary continuously with time. In this paper, we derive a closed-form analytical approximation for the first-passage time distribution of a single-boundary DDM with time-dependent drift, valid in the high-threshold regime. The main result allows us to analytically study the optimal timing of external signals to maximize the probability of a user response within our approximation framework. To evaluate our response time model, we conduct an extensive empirical comparison with state-of-the-art methods for user watch-time prediction and evaluation in simulated environments.

## 1. Introduction

The timing of human decisions reflects complex and partially hidden cognitive dynamics. From social media users deliberating when to share their content, to traders deciding when to execute a trade, appropriate response times constitute important human decisions. These response times encode both an agent's internal state and the information arriving from the external environment. Outside of controlled laboratory settings, decision-making rarely occurs in isolation; individuals are continuously exposed to asynchronous signals such as notifications, messages, or market updates. Modeling how such external information affects the timing of actions constitutes a growing field of interest, as it is

shown in a wide variety of applications (Holme & Saramäki, 2012; Hawkes, 2018; Khatuya et al., 2025).

A dominant statistical framework for modeling event timings is based on the literature of temporal point processes (Shchur et al., 2021), where actions are generated by a stochastic conditional intensity that depends on past events and covariates. Self-exciting models, most notably Hawkes Processes (Hawkes, 1971), have been widely used to capture clustering and excitation effects in social systems, finance, and online platforms (Rizoiu et al., 2017; Bacry et al., 2015). More recent work has substantially expanded the expressive power of point processes through neural parameterizations of the intensity function (the instantaneous probability rate of an event occurring in a point process, conditional on the history) (Du et al., 2016; Mei & Eisner, 2017; Shchur et al., 2021).

Using Deep Learning to model reaction times can lead to good prediction tools, but due to the generality of those methods, they can be inefficient to capture well known features of human behavior. For instance, consider Reinforcement Learning with JITAI (Just-in-time Adaptable Interventions). To find the optimal intervention policy, the RL algorithm can overwhelm users creating too many interventions in short periods of time (Wang et al., 2021), or create habituation (Liao et al., 2018). One strategy to overcome the fatigue problem is to impose frequency constraints, generating the problem of learning the constraint (Zhao et al., 2018). The same problem can also be seen as 'long-term valued' interactions with users (O'Brien et al., 2022).

While it is evident ex-post that elements, such as 'resting time' (refractory periods), are necessary, standard self-exciting models do not naturally account for inhibition. For instance, in a standard linear Hawkes Process, the kernel function is constrained to be non-negative to ensure the conditional intensity function remains valid. As a result, the model cannot naturally capture the temporary suppression of activity immediately following an event, necessitating the use of nonlinear extensions to incorporate such inhibitory effects (Costa et al., 2020).

Nevertheless, there is extensive literature providing insights on how to model decision timing from a cognitive per-

[1]Stern School of Business, New York University, New York, NY, USA. Correspondence to: Gustavo Grivol <ggs9313@stern.nyu.edu>, Alexander Tuzhilin <at2@stern.nyu.edu>, Hanna Halaburda <hh66@stern.nyu.edu>.

*Proceedings of the 43rd International Conference on Machine Learning*, Seoul, South Korea. PMLR 306, 2026. Copyright 2026 by the author(s).

spective, including a popular framework of *drift-diffusion modeling (DDM)*, in which noisy evidence accumulates until a decision boundary is reached (Wald, 1947; Stone, 1960; Ratcliff, 1978). The DDM framework has been very successful in explaining choice and response-time distributions across various domains, including perceptual decision-making (Ratcliff & McKoon, 2008; Shadlen & Kiani, 2013) and value-based choice (Fehr & Rangel, 2011; Krajbich et al., 2010; Krajbich & Rangel, 2011; Krajbich et al., 2012). Recent work has further characterized optimal time-varying decision boundaries that account for uncertainty and time costs, providing insight into the speed–accuracy tradeoff (Milosavljevic et al., 2010; Drugowitsch et al., 2012; Fudenberg et al., 2018; Tajima et al., 2019). Despite these advances, classical DDMs are typically studied in controlled experimental environments and are rarely used as generative models for real-world event streams.

Extending the DDM framework to settings with time-varying drift or boundaries, such as those induced by irregular external signals, poses severe tractability challenges (Smith, 2000). In general, the first-passage time distribution of a diffusion process to a time-dependent boundary does not admit a closed-form expression (Durbin, 1985; Salminen, 1988; Downes, 2008). This limitation is well documented in the applied probability literature on boundary-crossing problems (Lerche, 2013; Cuzick, 1981; Borovkov & Downes, 2010). While numerical methods can simulate such processes, the absence of analytical expressions for the first passage time density hinders likelihood-based inference and makes optimization or control problems computationally demanding (Shinn et al., 2020; Fengler et al., 2021). Existing approximation approaches, including piecewise-linear boundary methods or image-based techniques (Daniels, 1996; Smith & Ratcliff, 2022), either rely on restrictive assumptions or lack formal approximation guarantees.

A direct consequence of this intractability is the difficulty of computing likelihoods for response-time data under generalized DDMs. Recent work addresses this issue using numerical approximations (Shinn et al., 2020; Boehm et al., 2021) or neural likelihood estimators (Fengler et al., 2021; Boelts et al., 2022). While these methods substantially expand the class of models that can be estimated, they usually incur high computational costs and often have limited analytical insight in the density of the first passage time since the likelihood is obtained through black-box models (Fengler et al., 2021).

To better understand response time of users in multiple platforms, focusing exclusively on single choice problems, in this paper we consider the decision making method of the *Go/No-Go Drift-Diffusion model* that has been extensively studied before (Gomez et al., 2007). Our main result is obtaining a closed-form approximation of the density of the *Go/No-Go Drift-Diffusion model* under a single boundary model and a fairly general class of drift functions. Our approximation is suitable for high-threshold models, which can be seen as decisions that require prolonged evidence accumulation and exhibit long response times. For example, the timing between seeing a webpage of a product and clicking on it to buy the product. Crucially, the resulting approximation expression remains tractable even when the drift is a function of new variables arriving during the decision process, allowing external signals to be incorporated directly into the decision dynamics and yielding a well-defined likelihood over event times.

We later use this structure of response time to study analytically the timing of interventions that influence decision-making through their effect on the drift of the diffusion process. As a motivating application, we consider user reactions in digital platforms, where messages or content exposures act as external signals that may increase or decrease the probability of an action occurring within a given time window. By retaining an explicit representation of the decision process, our approach enables interpretable analysis of intervention timing, in contrast to the black-box control methods for point processes (Farajtabar et al., 2017; Upadhyay et al., 2018; Qu et al., 2023). More broadly, our results provide a theoretical foundation for modeling event timing under continuous information flow and for designing timing policies in settings where analytical tractability is essential.

We summarize the main contributions of this paper below:

1.We propose a novel model that is (a) grounded in a rigorous cognitive theory, (b) scalable across a wide range of applications and (c) interpretable: the model allows us to retain the interpretability inherited from the underlying theory. Note that scalability is possible because the proposed model can be estimated via likelihood-based methods using an approximate closed-form solution, as derived in this paper, which is especially evident when our model is compared to the simulation-based estimation methods that typically incur much higher computational costs.

2. The proposed closed-form solution enables us to provide novel theoretical results for explicitly determining the nudge timings within a theoretical cognitive decision model. This theoretical closed-form approach provides significant advantages over the previously proposed methods that rely on reduced-form point processes or on testing strategies empirically without prior behavioral model.

3. We empirically validate the proposed model on two real-world applications involving user decision timing and show that it significantly outperforms various baselines across multiple experimental settings and evaluation metrics. Beyond predictive performance, the model yields interpretable

insights into how time-varying external information shapes user behavior.

## 2. Background and Previous work

As mentioned previously, the decision process is inspired by the *Go/No-Go Drift-Diffusion model* (Gomez et al., 2007). In experimental settings, subjects are often instructed to respond only when a stimulus belongs to a target category, as in *Go/No-Go* lexical decision tasks where participants respond to words and withhold responses to nonwords (Ratcliff et al., 2004; 2018), or modeling decisions involving impatience or waiting costs (Shenoy & Yu, 2012). As an example, consider a user who decides to open an app for an activity, such as watching a new video. The user has a continuous decision process throughout the day representing her willingness to open the app, where the implicit decision process represents the evidence process, and opening the app represents the 'Go' action. During the decision process, the user receives notifications or similar reminders, which the *Go/No-Go Model* incorporates through the drift of the evidence process. When the willingness to act passes the threshold, then we observe a 'Go' action, and the willingness to act is reset back to zero right after this.

In the *Go/No-Go* tasks, the "No-Go" (not reacting) is typically modeled in one of the two ways: assuming an implicit boundary for the decision of not acting, or utilizing a single-boundary model (Gomez et al., 2007). For instance, in the app opening example, the two-boundary model would represent a user implicitly deciding that she would not open the app (but without taking any observable action). Although previous research shows that single-boundary models have good fit of the reaction time distribution (Trueblood et al., 2011), there is also evidence suggesting that a standard single boundary model (without time dependent drift) may fit experimental data poorly compared to a two boundary model (Gomez et al., 2007). To address this problem, we rely on flexible formulation for the drift process, as is shown in Section 5.4. Furthermore, the single-boundary formulation offers useful theoretical properties in our context, as it permits a non-zero probability of "no event" over an infinite horizon by allowing for constant negative drift. After each crossing, the process resets to level zero immediately, restarting the process as in (Nguyen et al., 2019).

### 2.1. Related work

Our work is related to three lines of research: (a) theoretical framework of Drift Diffusion Models (DMMs), (b) timings of nudges and (c) empirical applications of DMMs. We will review them in this section.

First, we made significant progress developing an estimation method for drift diffusion models presented in (Shinn et al., 2020; Fudenberg et al., 2020; Fengler et al., 2021). There is an interesting trade-off that our method presents compared to this literature: While our method depends on the characteristics of the decision to gain accuracy (high threshold decisions), which the previous methods do not depend, our method gain scalability and parametric flexibility since we estimate a closed form likelihood for a general drift function.

Second, in estimating the point process, models such as neural Hawkes Processes (Mei & Eisner, 2017) and transformer Hawkes models (Zuo et al., 2020), are very popular due to their general structure and few limitations. A similar case is for the papers predicting the time of user engagement using deep learning and machine learning methods (Zhao et al., 2025; Sun et al., 2024; Lin et al., 2023). Although these papers report strong performance results, they do not provide any theoretical backbone for modeling user behavior. In contrast to this prior work, in our case, we directly derived the decision time from the theoretical principles of decision making. Third, in Section 4 we study a problem mostly analyzed empirically: timing of nudges and interventions in real-world systems, including notification scheduling (O'Brien et al., 2022) and Just-in-Time Adaptive Interventions in mobile health (Liao et al., 2018).

## 3. Model

For a summary of the theoretical notation of the model, Table 4 is available in Appendix 1 for future reference.

Consider a single agent facing a sequential decision problem within a time window $[0, T]$. As the agent takes each action, such as buying multiple items in a website or liking posts in social media, it generates a sequence of event times $\tau^k$, $k = 1, 2, \ldots$, where $\tau^k \in [0, T]$. The actions only differ by their timing. In other words, we assume that there is a single type of action that the agent chooses when to take.

We denote the agent's willingness to take action by the evidence process $(Z_t)_{t \geq 0}$. The decision threshold is given by $u > 0$, representing the accumulated willingness necessary to trigger a signal. For example, representing the user's social cost to share a post. When $Z_t < u$, the process evolves as a standard Brownian motion plus a drift term $\mu(X(t), t)$ that depends on the external signal function $X(t)$. For the cost of nudging, the function $X(t)$ can represent a time-discounted summation of nudges received up to time $t$, as will be explained further in Section 4. This is described by the following Stochastic Differential Equation (SDE):

$$dZ_t = \mu(X(t), t)dt + dB_t, \quad \text{for } Z_t < u \text{ and } t \geq 0, \quad (1)$$

where $dB_t$ denotes standard Brownian motion. The agent generates a signal (takes action) whenever the process touches the threshold, i.e., $Z_t = u$. Regarding the drift term,

we impose two constraints: **A1.** The function $\mu(X(t), t)$ is Riemann integrable. **A2.** $|\mu(X(t), t)| < M$, for all $t \geq 0$.

*Remark* 3.1 (Unit diffusion as rescaling). For any constant diffusion coefficient $\sigma > 0$, the same first-passage time distribution can be obtained by rescaling: the process $Z_t/\sigma$ with threshold $u/\sigma$ and drift $\mu/\sigma$ yields an equivalent unit-diffusion model. Therefore, the diffusion magnitude is absorbed via rescaling of the drift and threshold parameters, both of which are estimated in our framework.

Immediately after sending a signal, the evidence process $Z_t$ resets to 0. Thus, if $t$ is such that $Z_t = u$, then $Z_{t+} = 0$. We define the sequence of hitting times $\tau_u^k$ associated with a task with threshold $u$, referred to as hereafter as *reaction times*:

$$\tau_u^k = \inf\{t : Z_t = u,\ t > \tau^{k-1}\}, \quad \text{for } k \in \mathbb{N}, \quad (2)$$

where we define $\tau^0 = 0$. We also use $\tau_t$ to denote the last hitting time prior to time $t$: $\tau_t := \sup\{\tau^k : \tau^k \leq t\}$.

To formally capture the probabilistic structure, we fix a probability space $(\Omega, \mathcal{F}, \mathbb{P})$ supporting a standard Brownian motion $B = (B_t)_{t \geq 0}$. Let $\mathbb{F} = (\mathcal{F}_t)_{t \geq 0}$ denote the natural filtration generated by $B$. The external signal is treated as a deterministic, Borel-measurable function $X : [0, \infty) \to \mathbb{R}$ (or $\mathbb{R}^K$). Consequently, the drift coefficient $\mu(X(t), t)$ is a deterministic function of time, and the process $Z$ is an $\mathbb{F}$-adapted Itô diffusion (Ratcliff & McKoon, 2008; Fudenberg et al., 2020). The reaction times $\tau_u^k$ are, therefore, $\mathbb{F}$-stopping times. Finally, we define the pair $\xi_t := (\tau_t, (X(s))_{s \in [0,t]})$ to simplify the notation in subsequent sections.

### 3.1. Main Model Approximation

The main theorem of this section approximates the counting process generated by $Z_t$ passing a high threshold $u$. High decision thresholds are associated with tasks that explicitly emphasize accuracy or impose high costs on errors, as well as tasks requiring cognitive control or response inhibition (Ratcliff & McKoon, 2008). As an example of how high thresholds can be associated with accuracy cost: In experiments where participants were asked to prioritize accuracy reliably, it was estimated a higher boundary separation (Ratcliff & Rouder, 1998; Bogacz et al., 2010). Similarly, when errors carry asymmetric penalties, participants adopt higher thresholds to avoid costly mistakes consistent with reward-maximizing policies (Bogacz et al., 2006).

The main theoretical result of this paper is presented in Theorem 3.1 that provides a closed-form approximation of the distribution of $\tau_u^k$ for large values of $u$.

**Theorem 3.2.** *Consider the following function:*

$$f(t, u, \xi_t) := \frac{u - \int_{\tau_t}^{t} \mu(X(s), s)ds}{(t - \tau_t)^{1/2}} \quad (3)$$

*Then, under assumptions (A1)-(A2):*

$$P(\tau_u^k \in [t, a] \mid \tau_t)$$
$$\approx \int_t^a \left[ \frac{f(s, u, \xi_s)}{2(s - \tau_t)} - f_t(s, u, \xi_s) \right] \phi(f(s, u, \xi_s))ds \quad (4)$$

*as $u \to \infty$. Where $\phi$ denotes the standard normal density and $f_t$ denotes the partial derivative with respect to time.*

The proof of this theorem can be found in the Appendix, and it follows from certain types of adjustments of the boundary crossing results from Cuzick (1981). Note that the context in which the closed-form approximation of the distribution of $\tau_u^k$ takes place, resembles most of the contexts in which agents take decisions on various platforms, such as deciding to buy an item, watch a video, or send a text message; but these tasks have high costs of making mistakes.

Next, we define a specific process based on this approximation:

**Definition 3.3.** We say that a sequence of stopping times $\{\bar{\tau}^k\}_{k=1}^{\infty}$ follows an Extended Drift Diffusion Model (ExtDDM) process with drift $\mu : \mathbb{R}^K \times \mathbb{R}_+ \to \mathbb{R}$, satisfying (A1) and (A2), threshold $u > 0$, and external function $X(t) \in \mathbb{R}^K, t \geq 0$, if its cumulative distribution function is given by:

$$P(\bar{\tau}^k \leq a \mid \bar{\tau}_t) = \quad (5)$$
$$\frac{1}{C_X^{\bar{\tau}}} \int_{\bar{\tau}_t}^{a} \left[ \frac{f(s, u, \xi_s)}{2(s - \bar{\tau}_t)} - f_t(s, u, \xi_s) \right]^+ \phi(f(s, u, \xi_s))ds$$

where the function $f(s, u, \xi_s)$ follows the definition in (3) with $\xi_t := (\bar{\tau}_t, (X(s))_{s \in [0,t]})$, and $[y]^+$ denotes $\max(0, y)$. The normalizing constant $C_X^{\tau^{k-1}}$ is defined as:

$$C_X^{\bar{\tau}_t} = \max \left\{ 1, \int_{\bar{\tau}_t}^{\infty} \left[ \frac{f(t, u, \xi_t)}{2(t - \bar{\tau}^{k-1})} - f_t(t, u, \xi_t) \right]^+ \right.$$
$$\left. \times \phi(f(t, u, \xi_t))dt \right\} \quad (6)$$

By comparing the CDF in this definition with the main theorem, we observe two adjustments: the normalizing constant and the restriction to the positive part of the integrand. Calculating $C_X^{\tau^{k-1}}$ may present computational challenges, and requires knowledge of the exogenous function ahead of time. The following proposition provides sufficient conditions to guarantee that $C_X^{\tau^{k-1}} = 1$, thereby avoiding the computation of the constant.

**Proposition 3.4.** *Consider an ExtDDM model with drift $\mu(t)$ such that: $u - \int_0^t \mu(s)ds + t\mu(t) > 0, \forall t > 0$. Then each of the following conditions is sufficient to guarantee $C_X^{\tau^{k-1}} = 1$:*

- *Condition 1: If $f(t, u, \xi_t) \to 0$ as $t \to \infty$ and*

$$\int_0^\infty \frac{\mu(t)}{\sqrt{2\pi t}} e^{-\left(u - \int_0^t \mu(s)ds\right)^2/(2t)} dt \geq 0 \qquad (7)$$

- *Condition 2: If $f(t, u, \xi_t) \to -\infty$ as $t \to \infty$ and*

$$\int_0^\infty \frac{\mu(t)}{\sqrt{2\pi} t^{3/2}} e^{-\left(u - \int_0^t \mu(s)ds\right)^2/(2t)} dt \geq 1 \qquad (8)$$

The integral condition in Condition 1 is easily satisfied by any positive drift, since the integrand is then non-negative. Note, however, that the convergence requirement $f(t, u, \xi_t) \to 0$ as $t \to \infty$ is not automatically satisfied by every positive drift: for a strictly positive constant drift, $f(t, u, \xi_t) = (u - \int_0^t \mu(s)ds)/\sqrt{t} \to -\infty$. The convergence condition requires the drift to grow at a rate compatible with $u - \int_0^t \mu(s)ds = o(\sqrt{t})$, which is satisfied, for instance, by positive drifts whose cumulative integral $\int_0^t \mu(s)ds$ approaches $u$ at a rate slower than $\sqrt{t}$, or by drifts that are positive over a finite interval $[0, a]$ where $a$ depends on the threshold $u$. The second condition gives us a similar characterization, but considering the case in which the $f(t, u, \xi_t)$ goes minus infinity, therefore covering cases where the cumulative drift grows faster than $\sqrt{t}$, for instance, when $\mu$ is bounded below by a positive constant.

## 4. Timing Optimization under Kernel Structure

In this section, we analyze a specific structure for the drift to study the timing decisions platforms face when sending advertisements, notifications, or interventions. The main goal is to relate the positioning of a signal to the high-threshold context.

Following an approach similar to the self-exciting structure of Hawkes processes, we consider a drift that is affected by signals $x_j \geq 0, j = 1, 2, \ldots$ received from the environment up to time $t$. The drift therefore depends on the history of events $H_t = \{x_j : x_j \leq t\}$. These signals impact the drift according to the following structure:

$$X(t) = \sum_{x_j \in H_t} h(t - x_j) \qquad (9)$$

$$\mu(X(t), t) = \Psi(X(t)) \qquad (10)$$

Where $\Psi : \mathbb{R} \to \mathbb{R}$ and $h : \mathbb{R} \to \mathbb{R}$ are both differentiable, $\Psi$ is bounded and we also assume $h(s) = 0$, for $s \leq 0$. These assumptions will guarantee the boundedness of the drift, as required by the approximation, and by constraining $h(s) = 0, s \leq 0$, we avoid accounting the nudge into the drift before its arrival time.

To analyze how this structure leads to different timing strategies, we introduce an objective function inspired by the

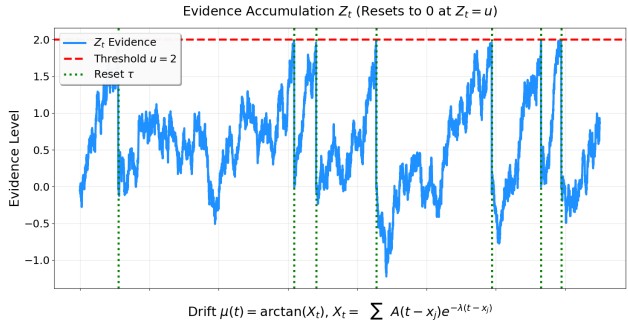

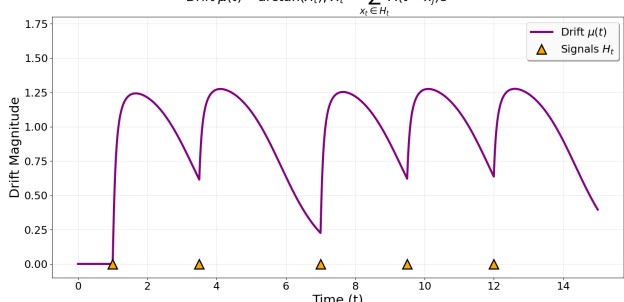

*Figure 1.* Sample realization of the Evidence Accumulation process $Z_t$ with resets (bottom) and the corresponding time-varying drift $\mu(t)$ (middle)

platforms optimizing for 'daily active users'. Since active user requires at least one action per time window, the associated objective can be written as a maximization of the hitting time c.d.f. under the time window of interest:

**Definition 4.1.** For an interval $[0, a]$ where $0 \leq a \leq \frac{u-1}{M}$, the objective $OPT^J_{[0,a]}$ is defined as:

$$\sup_{0 < x_1 \leq x_2 \leq \ldots \leq x_J \leq a} P(\bar{\tau}^1 \leq a) \qquad (OPT^J_{[0,a]})$$

Such objective can be found in app notifications studies, such as (Yancey & Settles, 2020). Below we provide the main result on how to compare nudge timing strategies under high thresholds:

**Theorem 4.2.** *Fix $a > 0$ and let $\{z_j\}_{j=1}^J, \{y_i\}_{i=1}^I \subset [0, a]$ be two finite signal sequences. Assume $\Psi : \mathbb{R} \to \mathbb{R}$ is strictly increasing and $h$ is measurable. Let $Z := \{z_j\}_{j=1}^J$ and $Y := \{y_i\}_{i=1}^I$, and let $P_Z$ and $P_Y$ denote the probability measures of the stopping time $\bar{\tau}^1$ under signal sequences $Z$ and $Y$, respectively.*

*If there exist $\varepsilon > 0$ and $c \in [0, a)$ such that*

$$\int_0^t \Psi\left(\sum_{j=1}^J h(s - z_j)\right) ds + \varepsilon \qquad (11)$$

$$< \int_0^t \Psi\left(\sum_{i=1}^I h(s - y_i)\right) ds \quad \text{for all } t \in [c, a],$$

*and if*

$$u > \max\left\{2Ma + \sqrt{(aM)^2 + \frac{3a}{2}}, \ a\left(\frac{2\ln 2}{\varepsilon} + M + \varepsilon\right)\right\},$$

(12)

*then $P_Y(\bar{\tau}^1 < a) > P_Z(\bar{\tau}^1 < a)$.*

Although this result does not provide direct optimality, it reveals a simple principle for establishing stochastic dominance between signal sequences: Condition (12) demonstrates that for sufficiently high thresholds, one only needs to compare the integral of the drift near the end of the interval to determine which signal sequence yields a higher probability of action. If we look at nudges that have vanishing effects, a simple way to position the nudge would be to have its peak effect right before the end of the time window. The next result reiterates this principle:

**Corollary 4.3.** *Fix $a > 0$ and suppose $\Psi : \mathbb{R} \to \mathbb{R}$ is strictly increasing. Consider the single-signal optimization problem $(OPT1_a)$.*

*(i) If $\Psi(h(s)) \geq 0$ for all $s \geq 0$, then $\lim_{u \to \infty} P_{x^1=0}(\tau^1 \leq a) = OPT^1_{[0,a]}$.*

*(ii) Suppose there exists $r > 0$ such that, for every $x \geq 0$, the mapping $s \mapsto \Psi(h(s - x))$ is nonnegative on $[0, x + r)$ and strictly negative on $(x + r, \infty)$. Then $\lim_{u \to \infty} P_{x^1=(a-r)}(\tau^1 \leq a) = OPT^1_{[0,a]}$.*

The proof of both cases follows directly from Theorem (12). Case (i) can be interpreted as signals permanently pushing the evidence process towards the boundary. In contrast, case (ii) describes a scenario with an initial excitation period (when $t < r$) followed by a refractory or inhibitory period.

In summary, in this section we demonstrated that, as we deal with the high-threshold scenarios, optimizing the time of nudges reduces to maximizing the total effect of nudges by the end of the time window. This effect is highly dependent on the functional form that the nudges will take into the drift, i.e. the $h$ and $\Psi$ functions, leading to variations where optimal nudge timing can be from the beginning of the time window or close to the end of the time window.

**Remark on multi-choice extension.** Note that in this paper we focused on a single action case. However, the framework presented above can naturally be extended to settings with multiple action types. Specifically, consider $J$ available choices indexed by $j \in S = \{1, \ldots, J\}$, where each action $j$ has its own evidence process and threshold. Under the assumption that the willingness-to-act processes for different options are mutually independent, then the probability that action $j \in S$ is selected can be obtained as the probability

of its corresponding exit time being the smallest:

$$P(Y = j) = \int_0^\infty f_j(t) \prod_{k \in S, k \neq j} (1 - F_k(t)) \, dt,$$

Where $Y$ represents the action being selected, $f_j$ and $F_j$ denote the pdf and cdf, respectively, of the first exit time for option $j$. We then can approximate the exit time distributions by ExtDDM under high thresholds. A full treatment of this extension, including the discussion of the independence assumption in relation to the prior literature, as in (Bogacz et al., 2006), is the subject of future research.

## 5. Experiments

We empirically validate the ExtDDM model now. Since the drift structure of our model is fairly general, in this section, we focus on an additional type of reaction time, i.e., the video viewing time in the context of watching online videos. Specifically, we predict when the user abandons watching a particular video. This prediction problem is an interesting application for our model since decision time is longer than for the traditional *Go/No-Go* case, which can indicate a higher decision threshold. Additionally, there are several models predicting user watching time that can be used as benchmarks in our study, therefore we can have a comparative assessment of our estimation.

Although empirical evaluation of the timing nudges studied in the previous section would be a natural addition to our empirical work, most datasets related to such user notifications and nudging face the following problems. First, to collect solid data on such notifications and nudges is very hard due to privacy considerations (Fraser et al., 2019) since full history of users with timestamps can easily lead to the identification of such users. Therefore, offsetting and rounding techniques are usually applied in such evaluations for privacy considerations, (Qu et al., 2025), which obscures the exact timing of user actions and makes such datasets inapplicable to our study. Therefore, we focus only on the video watching time prediction metric here, and describe it further below.

### 5.1. Watch-time Datasets

To demonstrate applicability of our method, we selected two datasets for estimating decision processes associated with the ExtDDM model. The first application focuses on video recommendations and uses the KuaiRec 2.0 dataset (Gao et al., 2022) containing approximately 12.3 million observations of users watched time for each short-video displayed on the KuaiRec platform. This dataset has the features pertaining to user demographics, video attributes, and historical viewing patterns.

In contrast to the short length video format of the KuaiRec

dataset, in the second application we explore longer video formats by predicting the watching time durations on the MOOCCubeX dataset from (Yu et al., 2021). The MOOCCubeX dataset contains millions of anonymized learner–resource interactions collected from the XuetangX platform, including fine-grained video watching logs. From these logs, the watch-time can be directly computed or aggregated at the video or session level. In addition, MOOC-CubeX provides contextual information on users and the videos, as will be later explored.

At Appendix B we conduct an ablation study over both empirical applications, showing the gains from a dynamic drift and threshold fixed over the sample.

### 5.2. Watch Time Prediction on KuaiRec

We applied the same pre-processing to the KuaiRec 2.0 watch time prediction model, as was done in (Zhao et al., 2025): we merged user and visual features along with the associated watching times for each user on each video. We applied a 10% downsample of the remaining entries.

The main challenge that we face in this application is the 'quick-skip' behavior, when users switch to the next video after watching the current one for a couple of seconds or even less (Sun et al., 2024; Zhao et al., 2025). The consequence of such behavior is high concentration of probability mass near zero watching time. Based on this, we identify two types of video watching behaviors that have been previously reported in the literature (Zhao et al., 2025): (a) users properly watching the video and subsequently deciding to stop engaging with it, and (b) users stop watching and deciding to switch immediately, thus hardly watching the video.

Since the 'quick-skip' behavior does not suit directly our decision making framework described at Section 3, we opted for a hybrid strategy. We create a 2-step model, where we first used a pre-trained EGMN model to predict if the video watching time would surpass the 66% quantile (in terms of the training data). If it does, then we used the ExtDDM model for the prediction purposes; and if it does not, then the prediction is based exclusively on the EGMN model (also being trained under this setting).

Denoting by $Z_{i,j}$ the features as processed by (Zhao et al., 2025), the ExtDDM architecture consisted of three components: We first optimized the encoder component $f_{enc} : \mathbb{R}^{36} \to \mathbb{R}^{64}$ taking video and user features and outputting a latent vector, where $f_{enc}$ is given by a 2-layer neural network with Layer Normalization. We then estimated another 3-layers neural network $f_{thr} : \mathbb{R}^{64} \to \mathbb{R}$, which was used to compute the threshold of the ExtDDM: $u(Z_{i,j}) = \ln(1 + e^{f_{thr}(Z_{i,j}) + u_{const}})$, $u_{const}$ being a positive constant coefficient. Lastly we compute the drift by first

creating a new vector of features $\bar{Z}_{i,j,t}$ using functions of the time elapsed from the beginning of the video and the latent vector $Z_{i,j}$: $\bar{Z}_{i,j,t} = [Z_{i,j}; \phi(t); Z_{i,j} \cdot t; Z_{i,j} \cdot (t+10^{-3})^{-1}]$, where $\phi(t)$ consisted of a polynomial expansion of $t$, an exponential expansion, and trigonometric functions of $t$. The model's loss was the negative log-likelihood, which was optimized over 20 epochs under Adam optimizer. For training and evaluation, we followed the split protocol used in the public EGMN implementation, which uses an 80%/20% train/test split without a separate validation set.

We compare our ExtDDM model with the following baselines in this paper:

**EGMN** (Exponential-Gaussian Mixture Network) (Zhao et al., 2025) Predicts parameters for a mixture distribution rather than a single value. The model optimizes the composite loss combining Negative Log-Likelihood (NLL) for distribution fitting, L1 loss for regression accuracy, and entropy regularization.

**CREAD** (Classification-Restoration Framework) (Sun et al., 2024). It discretizes the continuous watch time into adaptive buckets and its loss function combines Binary Cross Entropy for classification, Huber loss for the restored value, and an ordinal loss to ensure smooth transitions between buckets.

**TPM** (Tree-based Progressive Regression Model) (Lin et al., 2023). It breaks regression into a binary tree, where each node represents a conditional check, and then TPM predicts probabilities for every node, aggregating them to form the final prediction.

**VR** (Value Regression / Wide & Deep) (Cheng et al., 2016) model using the standard Wide & Deep architecture to perform direct point-estimation regression via Mean Squared Error (MSE). VR serves as a non-probabilistic baseline.

**CREAD+EGMN** To assess whether the gains of our 2-step framework stem from the ExtDDM formulation rather than from using a hybrid model, we evaluated an alternative hybrid model: CREAD+EGMN hybrid using the same 66% quantile split under the same 2-step framework detailed previously.

*Performance Metrics.* We used the following performance metrics in our study: (a) Mean Absolute Error (MAE) that is measured in seconds in our case; (b) XAUC (cross-user AUC), which assesses the model's ability to rank users by their expected watch times, a metric particularly relevant to recommender systems (Zhan et al., 2022); (c) KL-divergence (KL) that measures the distance between the histograms (100 bins) of the predicted watch-time and the real watch-time for the test sample.

Performance results are reported in Table 1. We can observe from these results that our 2-step model ExtDDM + EGMN significantly outperformed all other baselines in terms of

Table 1. Watch time regression performance

| Method | MAE | XAUC | KL |
|--------|-----|------|-----|
| VR | 4.6936 | 0.5383 | 1.4781 |
| EGMN | 4.4597 | 0.5805 | 0.8570 |
| TPM | 5.8098 | 0.5536 | 2.4928 |
| CREAD | 4.3863 | 0.5883 | 2.0255 |
| CREAD+EGMN | 4.4432 | 0.5753 | 0.3601 |
| ExtDDM + EGMN | **4.2833** | **0.6053** | **0.2063** |
|  | +2.3% | +16.1% | +42.1% |

Table 2. Watch time prediction on MOOCCubeX dataset

| Method | MAE | XAUC | KL |
|--------|-----|------|-----|
| EGMN | 37.181 | 0.5102 | 2.4245 |
| CREAD | 35.202 | 0.6321 | 4.3157 |
| ExtDDM | **32.342** | **0.6547** | **1.3453** |
|  | +8.1% | +17.1% | +44.5% |

the MAE, XAUC and KL-divergence metrics (percentages of performance improvements over the 2nd-best baseline are reported in Table 1). Note that, although the KL divergence does not measure predictive performance per se, it determines how close the distribution of the predicted values computed by our ExtDDM model is to the true distribution. Table 1 shows that ExtDDM + EGMN outperforms other baselines in terms of the KL divergence by a wide margin.

In Appendix B, we conduct an ablation study using the KuaiRec dataset, showing that the performance gains of the model decrease significantly if we remove the dynamic drift from our model.

## 5.3. MoocCubex

We also evaluated the proposed ExtDDM framework on the MOOCCubeX dataset, a large-scale collection of learning videos from the XuetangX MOOC platform. In contrast to the short video recommendation dataset, MOOCCubeX consists of long-form educational videos, making it well suited for modeling high-threshold, cognitively demanding engagement decisions.

The main set of dataset features was a vector of video embeddings generated using BERT, encoding video content into 384-dimensional embeddings by concatenating all subtitle text or using the video name if no subtitles were available. We also used feature enrollment patterns of the user (number of enrollments and enrollment window), and video features (video duration and start time, both normalized). We selected a 10% random subsample for the dataset. We split the data into 70% training, 10% validation, and 20% test sets.

ExtDDM is compared in this study to two strongest baselines of the last section, CREAD and EGMN, using identical feature sets and training splits. Since the 'quick-skip' behavior is not so prevalent for the MOOCCubeX dataset, we used the ExtDDM model directly, instead of the 2-stage model, as was done for the KuaiRec 2.0 implementation. The network architecture was expanded by increasing the size of the encoder, going from 2 layers to 4 layers. The threshold and drift architectures were kept the same.

The performance results are presented in Table 2 and show that ExtDDM dominates the EGMN and the CREAD models on the MOOCCubeX dataset in terms of the MAE, XAUC and KL metrics (performance improvements range from 8.1% for MAE to 17.1% for XAUC, discounting the 50% baseline).

## 5.4. Evaluations under alternative cognitive process

In this section, we evaluate recent model in deep learning for Temporal Point Processes in simulated environments where the data-generating mechanism follows structured cognitive decision processes. The goals of this evaluation are (a) to assess how models not grounded in cognitive theory perform when applied to the *Go/No-Go* style decision processes; and (b) to examine robustness and limitations of our approximation (as embodied in the ExtDDM model) when the true underlying decision dynamics depart from the assumptions of the proposed model.

We instantiated four distinct decision-making processes studied in the context of *Go/No-Go* process. The first is the Two-Boundary DDM (Implicit NoGo) where a lower boundary captures the implicit decision to withhold response (Gomez et al., 2007; Ratcliff et al., 2018), so at every crossing of the lower boundary the process resets without generating a new hitting time. Second is the Single-Boundary DDM, as described in Section 2, but using a constant drift. Third, is the Reflected Boundary DDM model that imposes a reflecting barrier at zero evidence, as described in (Hadian Rasanan & Amani Rad, 2021). Fourth, is the Collapsing Bounds (Time-Varying) model, where the drift rate decays dynamically to model strategic impatience, as studied in (Hawkins et al., 2015).

We benchmarked a set of deep learning architectures against these simulated environments. The baseline models, implemented via the EASYTPP library (Xue et al., 2023), included the Recurrent Marked TPP (RMTPP) and Neural Hawkes Process (NHP), which rely on recurrent history encoding to approximate the conditional intensity function, and the Intensity-Free model, which directly learns the interevent probability density. These were compared against our method ExtDDM, where we modeled the drift $\mu(t)$ via a polynomial-input feeding a neural network and then estimating the parameters under likelihood. The experimental

dataset consisted of 50,000 synthetic event sequences for each of the four *Go/No-Go* scenarios, generated with a time horizon of $T = 100$ and time-step $\Delta t = 0.01$. The data was stratified into training (60%), validation (20%), and testing (20%) sets. All models were trained via Maximum Likelihood Estimation (MLE) using the Adam optimizer with a learning rate of $10^{-3}$ and a batch size of 64 and we employed early stopping based on validation log-likelihood to prevent overfitting, with a patience parameter of 10 epochs.

*Table 3.* LogLikelihood Comparison

| Scenario | Model | $u = 4$ | $u = 8$ | $u = 12$ |
|---|---|---|---|---|
| Refl. | ExtDDM | -6.82 | -8.50 | -9.80 |
| | RMTPP | -17.41 | -320.42 | -530.45 |
| | NHP | -9.30 | -21.78 | -29.44 |
| | IntFree | -6.74 | -9.26 | -10.95 |
| 2-Bound | ExtDDM | -6.97 | -8.72 | -9.90 |
| | RMTPP | -33.70 | -365.12 | -571.59 |
| | NHP | -10.93 | -23.12 | -29.94 |
| | IntFree | -7.14 | -9.48 | -11.11 |
| Urgency | ExtDDM | -7.83 | -9.23 | -9.93 |
| | RMTPP | -459.91 | -773.21 | -582.79 |
| | NHP | -16.47 | -26.21 | -30.20 |
| | IntFree | -7.72 | -9.50 | -11.26 |
| 1-Bound | ExtDDM | -7.53 | -8.91 | -10.02 |
| | RMTPP | -271.51 | -524.20 | -663.40 |
| | NHP | -14.37 | -24.23 | -30.65 |
| | IntFree | -7.49 | -9.42 | -11.03 |

As observed from Table 3, ExtDDM has considerable gain in terms of Log-Likelihood for $u = 8.0$ and $u = 12.0$, but not for the low threshold scenario $u = 4.0$. Additionally, we observe that only the Intensity Free (Shchur et al., 2019) had a performance close to the ExtDDM. These results confirm the aforementioned utility of our ExtDDM method when it performs the best in the *high-threshold* applications (when the values of $u$ are high).

## 6. Conclusion

In this paper, we have proposed a novel model for estimating response times based on a closed-form approximation of the first-passage time under time-varying drift. Our closed-form approximation allows us to bridge the gap between rigorous cognitive theory and scalable methods. Unlike black-box neural point processes or simulation-heavy methods, our approach enables efficient, likelihood-based inference while retaining characteristics of human decision-making.

Leveraging our closed-form solution, we derive explicit theoretical conditions for optimal nudge timing, offering a rigorous advantage over heuristic strategies or reduced-

form point processes that lack behavioral priors. Empirically, we validate the model on two large-scale real-world applications, demonstrating that our proposed method significantly outperforms several DL-based baselines. Besides predictive gains, this structural approach yields interpretable insights into the decision making process, confirming that cognitive theory can be effectively scaled to complex, high-dimensional contexts.

## Impact Statement

This research introduces the Extended Drift-Diffusion Model (ExtDDM) to predict user decision times influenced by external signals, with applications in online nudges and recommender systems. While ExtDDM's high interpretability can help designers appropriately pace helpful notifications and optimize the delivery of information, the model should be applied in contexts that prioritize long-term user well-being.

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

# A. Proofs

*Table 4.* Table of Notation

| Symbol | Description |
|--------|-------------|
| **Decision Process** | |
| $Z_t$ | Evidence accumulation process at time $t$ |
| $u$ | Decision threshold (boundary) |
| $\mu(X(t), t)$ | Drift function depending on external signals |
| $B_t$ | Standard Brownian motion |
| $\tau_u^k$ | $k$-th reaction time (hitting time) |
| $\tau_t$ | Last hitting time prior to $t$ ($\sup\{\tau^k : \tau^k \leq t\}$) |
| **Approximation & ExtDDM** | |
| $\xi_t$ | History pair $(\tau_t, (X(s))_{s \in [0,t]})$ |
| $f(t, u, \xi_t)$ | Auxiliary approximation function (Eq. 3) |
| $\phi(\cdot)$ | Standard normal density function |
| $C_X^{\bar{\tau}}$ | Normalizing constant for ExtDDM |
| $\bar{\tau}^k$ | Stopping times following ExtDDM |
| **Timing & External Signals** | |
| $X(t)$ | External signal function |
| $x_j$ | Individual signal event times |
| $H_t$ | History of signals $\{x_j : x_j \leq t\}$ |
| $h(\cdot)$ | Kernel function for signal impact |
| $\Psi(\cdot)$ | Drift transformation function |

*Proof of Theorem 3.1.* Fix $0 \leq t < a$ and let $\tau_t$ denote the last hitting time of $u$ before $t$. By definition, the next hitting time $\tau_u^k$ falls in $[t, a]$ if and only if the evidence process $Z$ crosses $u$ in that interval. Conditioned on $\tau_t$, the post–threshold process on $[\tau_t, a]$ satisfies the SDE

$$dZ_s = \mu\big(X(s), s\big)\, ds + dB_s, \quad s \in [\tau_t, a], \qquad Z_{\tau_t} = 0,$$

so that for $s \in [\tau_t, a]$ one may write $Z_s = B_s - B_{\tau_t} + \int_{\tau_t}^s \mu(X(r), r)\, dr$. Defining the *curved boundary*

$$U_u(s) \;=\; u - \int_{\tau_t}^s \mu\big(X(r), r\big)\, dr$$

transforms the event $\{\sup_{s \in [t,a]} Z_s \geq u\}$ into a first–passage event for standard Brownian motion:

$$\Big\{ \sup_{s \in [t,a]} Z_s \geq u \Big\} \;=\; \Big\{ \sup_{s \in [t,a]} (B_s - B_{\tau_t}) \geq U_u(s) \Big\}.$$

For large thresholds $u$ the boundary $U_u$ diverges uniformly, and Cuzick's boundary–crossing theorem for Brownian motion gives the asymptotic

$$\mathbb{P}\left( \sup_{s \in [\tau_t, a]} Z_s \geq u \right) \sim \int_{\tau_t}^a \left[ \frac{f(s, u, \xi_s)}{2(s - \tau_t)} - \partial_s f(s, u, \xi_s) \right] \varphi\big(f(s, u, \xi_s)\big)\, ds, \qquad (13)$$

where

$$f(s, u, \xi_s) \;=\; \frac{u - \int_{\tau_t}^s \mu(X(r), r)\, dr}{\sqrt{s - \tau_t}}, \qquad \phi(x) = \frac{1}{\sqrt{2\pi}} \, e^{-x^2/2},$$

and $\xi_s = (\tau_t, X|_{[\tau_t, s]})$. This result holds when $\mu$ is Riemann integrable and bounded (assumptions (A1)–(A2)), because these hypotheses ensure that the boundary $U_u$ is differentiable and grows fast enough to satisfy the conditions of Cuzick's theorem.

It remains to relate (13) to the distribution of $\tau_u^k$. Since $\tau_u^k \leq x$ is equivalent to $\sup_{s \in [\tau_t, x]} Z_s \geq u$, differentiating the crossing probability in (13) with respect to the upper limit shows that the integrand in (13) approximates the density of $\tau_u^k$. Under the boundedness of $\mu$, a short calculation shows

$$\partial_s f(s, u, \xi_s) = -\frac{\mu(X(s), s)}{\sqrt{s - \tau_t}} - \frac{f(s, u, \xi_s)}{2(s - \tau_t)},$$

and hence for large $u$ the term $\partial_s f$ is negative. Substituting this derivative into the integrand yields

$$f'_{(\tau_t, u_i, X_t)}(t) = -\frac{\mu(X_t, t)}{(t - \tau_t)^{1/2}} - \frac{u_i - \int_{\tau_t}^t \mu(X_t, t)ds}{2(t - \tau_t)^{3/2}} \tag{14}$$

$$\leq \frac{M}{(t - \tau_t)^{1/2}} - \frac{u}{2(t - \tau_t)^{3/2}} + \frac{(t - \tau_t)M}{2(t - \tau_t)^{3/2}}$$

For a constant drift $\mu$, the right–hand side reduces to $\frac{u}{(s - \tau_t)^{3/2}}$, and the integral in (13) reproduces the exact first–passage–time density of the classical drift–diffusion model. Consequently,

$$\mathbb{P}\big(\tau_u^k \in [t, a] \,\big|\, \tau_t\big) = \mathbb{P}\big(\tau_u^k \leq a \,\big|\, \tau_t\big) - \mathbb{P}\big(\tau_u^k \leq t \,\big|\, \tau_t\big) \approx \int_t^a \left[\frac{f(s, u, \xi_s)}{2(s - \tau_t)} - \partial_s f(s, u, \xi_s)\right] \varphi\big(f(s, u, \xi_s)\big) \, ds,$$

which proves the approximation stated. $\qquad\square$

**Proposition 3.3:**

*Proof.* 1. To determine if $C = 1$, we need to have the following integral lower than one:

$$I = \int_0^\infty \frac{u - \int_0^t \mu(s)ds + t\mu(t)}{t^{3/2}} e^{-\left(u - \int_0^t \mu(s)ds\right)^2/(2t)} \frac{1}{\sqrt{2\pi}} dt \tag{15}$$

We can decompose the integrand by relating it to the derivative of the complementary cumulative distribution function (CDF) of the standard normal distribution:

$$\Phi^*(x) = \frac{1}{\sqrt{2\pi}} \int_x^\infty e^{-z^2/2} dz$$

Let us now compute the derivative of the composite function $\Phi^*\big(f_{\tau, X_t}(t)\big)$ with respect to $t$ using the chain rule:

$$\frac{d}{dt} \Phi^*\big(f_{\tau, X_t}(t)\big) = \Phi^{*\prime}\big(f_{\tau, X_t}(t)\big) \cdot \frac{d}{dt} f_{\tau, X_t}(t)$$

$$= -\frac{1}{\sqrt{2\pi}} e^{-f_{\tau, X_t}(t)^2/2} \left(-\frac{\mu(t)}{\sqrt{t}} - \frac{u - \int_0^t \mu(s)ds}{2t^{3/2}}\right)$$

$$= \frac{1}{\sqrt{2\pi}} e^{-f_{\tau, X_t}(t)^2/2} \left(\frac{\mu(t)}{\sqrt{t}} + \frac{u - \int_0^t \mu(s)ds}{2t^{3/2}}\right)$$

The original integrand can be rewritten as:

$$\frac{u - \int_0^t \mu(s)ds + t\mu(t)}{\sqrt{2\pi}t^{3/2}} e^{-\left(u - \int_0^t \mu(s)ds\right)^2/(2t)} = 2\frac{d}{dt} \Phi^*\big(f_{\tau, X_t}(t)\big) - \frac{\mu(t)}{\sqrt{2\pi t}} e^{-f_{\tau, X_t}(t)^2/2}$$

By substituting this decomposition back into the integral for $I$, we get:

$$I = 2\left[\Phi^*\big(f_{\tau, X_t}(t)\big)\right]_0^\infty - \int_0^\infty \frac{\mu(t)}{\sqrt{2\pi t}} e^{-\left(u - \int_0^t \mu(s)ds\right)^2/(2t)} dt$$

1. At the lower limit, as $t \to 0^+$, then $f_{\tau,X_t}(t) \to +\infty$, so $\Phi^*(+\infty) = 0$. At the upper limit, if $f_{\tau,X_t}(t) \to 0$ as $t \to \infty$, then $\Phi^*(0) = \frac{1}{2}$. Therefore:

$$2\left[\Phi^*\left(f_{\tau,X_t}(t)\right)\right]_0^\infty = 2\left(\frac{1}{2} - 0\right) = 1$$

Hence we have:

$$I = 1 - \int_0^\infty \frac{\mu(t)}{\sqrt{2\pi t}} e^{-\left(u - \int_0^t \mu(s)ds\right)^2/(2t)} dt \leq 1 \tag{16}$$

2. This case differentiates from the previous by having $f_{\tau,X_t}(t) \to -\infty$ as $t \to \infty$. These limits give us $2\left[\Phi^*\left(f_{\tau,X_t}(t)\right)\right]_0^\infty = 2$, so the integral over the distribution is now given by:

$$I = 2 - \int_0^\infty \frac{\mu(t)}{\sqrt{2\pi t}} e^{-\left(u - \int_0^t \mu(s)ds\right)^2/(2t)} dt \tag{17}$$

Therefore leading to the integral condition in proposition.

$\square$

**Lemma A.1.** *Consider a differentiable and non-negative function $z : \mathbb{R}_+ \to \mathbb{R}$ such that $|z'(t)| < M, t \in \mathbb{R}_+$ and $z(0) = 0$, and define the following function for $u > 0$ and $\beta > 0$:*

$$\gamma_{u,\beta}(t) = \frac{e^{-(u-z(t))^2(2t)^{-1}}}{t^\beta \sqrt{2\pi}} \tag{18}$$

*Then for an interval $[0, a]$ and $u > 2Ma + \sqrt{(aM)^2 + a\beta}$, $\gamma_u(t)$ is strictly increasing over the interval $[0, a]$.*

*Proof of Theorem 4.2.* To simplify notation, define $z(t) = \int_0^t \Psi(\sum_{i=1}^J h(s - z_i)) ds$, then comparing the probabilities we have:

$$P_Z(\bar{\tau}^1 < a) - P_Y(\bar{\tau}^1 < a) \geq \int_c^a \phi\left(\frac{u - z(t) - \varepsilon}{t^{1/2}}\right)\left(\frac{u - z(t) + tz'(t) - \varepsilon}{t^{3/2}}\right) dt \tag{19}$$

$$- \int_0^a \phi\left(\frac{u - z(t)}{t^{1/2}}\right)\left(\frac{u - z(t) - tz'(t)}{t^{3/2}}\right) dt \tag{20}$$

$$= \int_c^a \frac{e^{-\frac{(u-z(t))^2}{2t}}}{\sqrt{2\pi}} e^{\frac{\varepsilon(u-z(t)-\varepsilon)}{2t}}\left(\frac{u - z(t) - tz'(t) - \varepsilon}{t^{3/2}}\right) dt$$

$$- \int_c^a \frac{e^{-\frac{(u-z(t))^2}{2t}}}{\sqrt{2\pi}}\left(\frac{u - z(t) - tz'(t)}{t^{3/2}}\right) dt$$

$$- \int_0^c \frac{e^{-\frac{(u-z(t))^2}{2t}}}{\sqrt{2\pi}}\left(\frac{u - z(t) - tz'(t)}{t^{3/2}}\right) dt$$

Now defining $k := \inf_{s \in [0,a]} z(s) - sz'(s)$, $K := \sup_{s \in [0,a]} z(s) - sz'(s)$ and $\gamma_u := \frac{e^{-\frac{(u-z(t))^2}{2t}}}{t^{3/2}\sqrt{2\pi}}$, we obtain the following inequality:

$$P_Z(\bar{\tau}^1 < a) - P_Y(\bar{\tau}^1 < a) \geq \int_c^a \gamma_u(t) e^{\frac{\varepsilon(u - \sup_{s \in [c,a]} z(s) - \varepsilon)}{2a}} (u - K - \varepsilon)dt \tag{21}$$

$$- \int_c^a \gamma_u(t)(u - k)dt - \int_0^c \gamma_u(t)(u - k)dt$$

$$= \int_c^a \gamma_u(t) \left[ e^{\frac{\varepsilon(u - \sup_{s \in [c,a]} z(s) - \varepsilon)}{2a}} (u - K - \varepsilon) - (u - k) \right] dt$$

$$- \int_0^c \gamma_u(t)(u - k)dt$$

$$\geq \gamma_u(c) \left[ (a - c) \left[ e^{\frac{\varepsilon(u - \sup_{s \in [c,a]} z(s) - \varepsilon)}{2a}} (u - K - \varepsilon) - u \right] - c(u - k) \right]$$

From the condition stated in the theorem the first integrand is positive, therefore we can use (18) to further simplify it:

$$P_Z(\bar{\tau}^1 < a) - P_Y(\bar{\tau}^1 < a) \geq \gamma_u(c) \left[ (a - c) \left[ e^{\frac{\varepsilon(u - \sup_{s \in [c,a]} z(s) - \varepsilon)}{2a}} (u - K - \varepsilon) - u \right] - c(u - k) \right] \tag{22}$$

We bound the difference of probabilities from below by

$$\mathbb{P}_Z(\bar{\tau}_1 < a) - \mathbb{P}_Y(\bar{\tau}_1 < a) \geq \gamma_u(c) \left[ (a - c) \left( e^{\frac{\varepsilon(u - \sup_{s \in [c,a]} z(s) - \varepsilon)}{2a}} (u - K - \varepsilon) - u \right) - c(u - k) \right]. \tag{23}$$

Since $\gamma_u(c) > 0$, it suffices to show that the expression inside the brackets is positive. A sufficient condition is

$$e^{\frac{\varepsilon(u - K - \varepsilon)}{2a}} (u - K - \varepsilon) > 2(u - k), \tag{24}$$

which implies

$$(a - c) \left( e^{\frac{\varepsilon(u - K - \varepsilon)}{2a}} (u - K - \varepsilon) - u + k \right) > c(u - k). \tag{25}$$

And using $|z'(t)| \leq M$ we obtain

$$k \geq -aM, \qquad K \leq aM.$$

Hence

$$\frac{2(u - k)}{u - K - \varepsilon} \leq \frac{2(u + aM)}{u - aM - \varepsilon}.$$

Therefore, condition (24) is satisfied whenever

$$e^{\frac{\varepsilon(u - aM - \varepsilon)}{2a}} > 2,$$

which is equivalent to

$$u > a \left( \frac{2 \ln 2}{\varepsilon} + M + \varepsilon \right).$$

$\square$

## B. Ablation Study

To isolate the contributions of specific components within the ExtDDM and ExtDDM+EGMN frameworks and identify the drivers of performance, we start evaluating the following variants of the ExtDDM+EGMN model on the KuaiRec dataset:

- **ExtDDM (Time-Invariant Drift)+ EGMN :** We removed the time-dependent features from the drift network inputs. This constrains the model to learn a constant drift rate $\mu$ rather than a dynamic, time-varying trajectory.

- **ExtDDM (Fixed Threshold)+ EGMN :** In this variant, we removed the threshold estimation network $f_{thr}(Z_{i,j})$. Instead of a personalized threshold, the model learns a single, global scalar parameter $u$ for the entire population.

- **ExtDDM+EGMN (Frozen):** This version initializes the EGMN component with pre-trained weights which are kept frozen (not updated) during the training of the hybrid model. This evaluates the benefit of fine-tuning the classifier within the joint framework.

- **ExtDDM+EGMN (Joint Training Only):** This model initializes the EGMN network randomly and trains it simultaneously with the ExtDDM component (without pre-training). This tests whether the decoupled pre-training of the classifier is necessary for convergence.

*Table 5.* Ablation study results on the KuaiRec dataset

| Model Variant | MAE | XAUC | KL |
|---|---|---|---|
| ExtDDM (Time-Invariant Drift)+ EGMN | 6.469 | 0.596 | 0.497 |
| ExtDDM (Fixed Threshold) + EGMN | 5.810 | 0.583 | 0.129 |
| ExtDDM + EGMN (Frozen) | 4.462 | 0.596 | 0.471 |
| ExtDDM + EGMN (Joint Training only) | 5.148 | 0.560 | 0.333 |
| **Full ExtDDM + EGMN** | **4.283** | **0.605** | **0.206** |

The ablation study on the KuaiRec dataset demonstrates that the implementation of dynamic drift is a critical driver of the model's predictive accuracy; the "Time-Invariant Drift" variant, which replaced time-dependent features with a constant rate, saw a substantial increase in Mean Absolute Error (MAE) from 4.283 to 6.469, confirming that a dynamic, time-varying trajectory is essential for effectively modeling user watch time. We also observe decrease in performance for the 'Joint Training only' model. It is important to note that the EGMN component performs a dual role in this framework: it functions as the predictor for the second step when the predicted watch time is low, while also serving as the decision-maker that determines whether a sample is handled by the EGMN itself or passed to the ExtDDM for prediction.

Lastly, we perform similar analysis at table 6. The table shows similar interpretation, while both components have high impact in deterioration of MAE and XAUC, the time variation of the drift seems to play major role in capturing dynamics of engagement.

*Table 6.* Ablation Study on MOOCCubeX Dataset

| Model Variant | MAE | XAUC | KL |
|---|---|---|---|
| ExtDDM (Time-Invariant Drift) | 39.52 | 0.512 | 2.158 |
| ExtDDM (Fixed Threshold) | 35.81 | 0.581 | 1.842 |
| **Full ExtDDM** | **32.34** | **0.655** | **1.345** |

## C. Threshold-MAE Analysis

To assess the practical validity of the high-threshold assumption on our real-world experiments, we analyzed the relationship between the estimated thresholds and the predictive performance of ExtDDM and the EGMN baseline on both KuaiRec and MOOCCubeX. We used 244,431 test samples and measured the threshold–MAE relationship via two regressions: (a) a linear regression of MAE against the threshold $u$, and (b) a regression of $\log(\text{MAE}_{\text{EGMN}}/\text{MAE}_{\text{ExtDDM}})$ against $u$. The latter directly captures whether ExtDDM's relative advantage grows with the threshold.

*Table 7.* MAE–threshold regression on KuaiRec

| Metric | Coefficient on $u$ | $p$-value |
|---|---|---|
| $\text{MAE}_{\text{ExtDDM}}$ | $-0.61$ | 0.252 |
| $\log(\text{MAE}_{\text{EGMN}}/\text{MAE}_{\text{ExtDDM}})$ | 0.93 | 0.000 |

*Table 8.* MAE–threshold regression on MOOCCubeX

| Metric | Coefficient on $u$ | $p$-value |
|---|---|---|
| $\text{MAE}_{\text{ExtDDM}}$ | 18.93 | 0.070 |
| $\log(\text{MAE}_{\text{EGMN}}/\text{MAE}_{\text{ExtDDM}})$ | 0.4364 | 0.000 |

The MAE decreases (modestly) with the threshold for KuaiRec, while the opposite holds for MOOCCubeX. However, both first-row coefficients are not significant at the 5% level. In contrast, the comparative regressions (second rows) show that EGMN's MAE grows relative to ExtDDM's as the threshold grows, and this is statistically significant for both datasets. This supports the theoretical prediction that ExtDDM's relative advantage increases in the high-threshold regime.

The threshold distribution statistics across the 244,431 test samples are reported in Table 9.

*Table 9.* Distribution of the estimated thresholds across test samples

| Variable | Min | Max | Mean |
|---|---|---|---|
| KuaiRec (normalized) | 1.684 | 4.04 | 3.54 |
| KuaiRec (seconds) | 4.339 | 4.51 | 4.42 |
| MOOCCubeX (normalized) | 1.44 | 3.34 | 2.23 |
| MOOCCubeX (seconds) | 3.12 | 3.31 | 3.29 |

The estimated thresholds fall in a range that broadly matches the $u = 4$ column of Table 3, where the ExtDDM approximation already performs competitively.

## D. Nudge Timing Simulation

Because the available real-world datasets do not provide the dynamic nudge timestamps needed for a fair empirical evaluation of the nudge-timing theory (Section 4), we conducted a calibrated semi-synthetic simulation. We simulated 100,000 Brownian motion paths under both (a) the exact diffusion process and (b) the ExtDDM approximation, with drifts depending on a single nudge $x_1^* \in [0, 1]$ following the kernel structure of Equations (10)–(11) of the main text.

**Setup.** We considered two kernel families:

- *Permanent gain* $(i = 1)$: $h_1(s) = 5\exp(-s)$, $s \geq 0$.

- *Excitation followed by inhibition* $(i = 2)$: $h_2(s) = 5(0.3 - s)\exp(-s)$, $s \geq 0$.

We used $\Psi_i(x) = (M_i/\pi)\arctan(x)$, $i = 1, 2$, over the time interval $[0, 1]$, with a single nudge. To match the KuaiRec setting, we calibrated $M_1$ and $M_2$ so that the average absolute value of $\Psi_i$ over the time window matches the KuaiRec experiment, yielding $M_1 = 4.7748$ and $M_2 = 14.432$. We used the threshold values $u \in \{1.5, 2.0, 2.5, 3.0, 3.5, 4.0, 4.5\}$.

As the threshold grows, both the exact-diffusion and the ExtDDM-based simulations converge to the same qualitative regimes predicted by Corollary 4.3: the optimal nudge time concentrates at $x_1^* = 0$ in the permanent-gain case (signals permanently push toward the boundary, so earlier is better) and approaches $x_1^* \approx 0.7$ in the excitation–inhibition case (the optimal time is placed so that the peak of the kernel hits just before the end of the time window). The slight discrepancy between Tables 10 and 11 at moderate thresholds is consistent with the high-threshold validity regime of the approximation.

*Table 10.* Optimal single-nudge time $x_1^*$ under the exact diffusion

| Threshold $u$ | $x_1^*$ (Permanent) | $x_1^*$ (Exc-Inh) |
|---|---|---|
| 1.5 | 0.0 | 0.01 |
| 2.0 | 0.0 | 0.01 |
| 2.5 | 0.0 | 0.58 |
| 3.0 | 0.0 | 0.63 |
| 3.5 | 0.0 | 0.63 |
| 4.0 | 0.0 | 0.64 |
| 4.5 | 0.0 | 0.65 |

*Table 11.* Optimal single-nudge time $x_1^*$ under the ExtDDM simulation

| Threshold $u$ | $x_1^*$ (Permanent) | $x_1^*$ (Exc-Inh) |
|---|---|---|
| 1.0 | 0.3 | 0.0 |
| 1.5 | 0.0 | 0.0 |
| 2.0 | 0.0 | 0.7 |
| 2.5 | 0.0 | 0.7 |
| 3.0 | 0.0 | 0.7 |
| 3.5 | 0.0 | 0.7 |
| 4.0 | 0.0 | 0.7 |
| 4.5 | 0.0 | 0.7 |

## E. Drift Function Descriptive Statistics

To better understand the behavior of the drift function learned by ExtDDM on KuaiRec, we computed descriptive statistics across all test samples over the maximum time window:

*Table 12.* Descriptive statistics of the learned drift $\mu(X(t), t)$ on KuaiRec

| Metric | Value |
|---|---|
| Mean (all users, all periods) | 1.959411 |
| Max (all users, all periods) | 3.338117 |
| Min (all users, all periods) | 0.239779 |
| Std over time window (per user, averaged across users) | 0.010664 |
| Std over users (averaged over time window) | 0.002586 |

The minimum drift value across all users and time points is strictly positive, indicating that the learned drift consistently pushes the evidence process toward the decision boundary. The standard deviation over the time window (per user) is substantially larger than the standard deviation across users (per time point), which reinforces the empirical gains attributable to the dynamic, time-varying drift specification rather than to sample-specific heterogeneity alone.

