# OpenReview forum: "Approximating Drift-Diffusion Models for User Decisions under Nudging and External Information"
_ICML.cc/2026/Conference — ICML 2026 regular_

### Official Review · Reviewer_N91z · 2026-03-04

**Soundness:** 3
**Presentation:** 3
**Significance:** 3
**Originality:** 3
**Overall Recommendation:** 4
**Confidence:** 3

**Summary:**

This paper studies user decision times under external information using a drift-diffusion model with time-varying drift. Its main contribution is a high-threshold approximation to the first-passage-time distribution, which enables a tractable likelihood-based model (ExtDDM) with learnable drift and personalized thresholds. The paper combines this theory with experiments on synthetic settings and two watch-time datasets, where ExtDDM variants outperform several baselines, especially in higher-threshold regimes.

**Compliance With Llm Reviewing Policy:**

Affirmed.

**Final Justification:**

The rebuttal addresses my concerns, I therefore keep my positive score.

**Key Questions For Authors:**

1. Can you provide an end-to-end evaluation of the nudge-timing results, even in a semi-synthetic setting?
2. Can you report variance across runs, confidence intervals, and exact train/validation/test split protocols for KuaiRec and MOOCCubeX?
3. Can you clarify whether unit diffusion is a modeling assumption or a rescaling choice, and more explicitly connect the theorem quantities to the neural parameterization used in ExtDDM?

**Limitations:**

See Weaknesses and Questions.

**Strengths And Weaknesses:**

**Strengths**
- The paper clearly frames user decision duration as a drift-diffusion process influenced by external information and nudges, and explicitly defines the first-passage stopping-time target. This helps align the theory, likelihood construction, and prediction objective.
- The main theoretical result gives an explicit high-threshold approximation to the conditional crossing probability, replacing an otherwise intractable hitting-time law with a computable expression.
- The model is evaluated on both synthetic and real datasets, and the reported results on KuaiRec 2.0 and MOOCCubeX show improvements over multiple baselines. The appendix ablations are also useful because they isolate the roles of dynamic drift and learned thresholds.

**Weaknesses**
1. Although Section 4 develops a nudge-timing objective and theory, the experiments only test duration prediction, not whether the derived policy actually improves intervention outcomes.
2. The evaluation reports point estimates only, without standard deviations, confidence intervals, or multi-seed variability. The KL metric also depends on histogram binning, but no sensitivity analysis is provided.
3. The paper uses unit diffusion without clearly explaining whether this is a modeling restriction or a rescaling choice. Also, the main theorem depends on assumptions placed in the appendix, and the connection between theorem quantities and learnable model components could be explained more directly.
4. The manuscript does not clearly provide compute/resource details or a code/artifact statement, which makes reproduction harder.

---

> ### Author Rebuttal · Authors · 2026-03-30
>
> Thank you for your thoughtful questions, comments and suggestions. Our answers and additional results are provided below.
>
> ## Question 1:
>
> To provide an end-to-end evaluation of the nudge-timing results, we performed a simulation of $100,000$ Brownian motion paths with drifts depending on the nudges, as in the DDM with additional drift structure as in the model. To make our simulation more realistic, we calibrated the threshold and the drift to match the average behavior of the KauaiRec estimated model as will be described later. Moreover, we have also analyzed the nudge timing under the specified ExtDDM distribution. Both of these setups (including its drift structure as specified in equations (9) and (10) of the manuscript) are described below:
>
> - Permanent gain ($i=1$): $h_1(s)=5\exp(-s), s\geq 0$
> - Excitation followed by inhibition ($i=2$): $h_2(s)=5(0.3-s)\exp(-s), s\geq 0$
> - $\Psi_i(x)=M_i/\pi \cdot \arctan(x), i=1,2$
> - Time interval: $[0,1]$; Single nudge
>
> To realize similar settings, we matched the magnitude of the parameters found in the KuaiRec experiment. We ran the simulation with threshold values containing the intervals found across samples: $u\in\\{1.5, 2.0, 2.5, 3.0, 3.5, 4.0, 4.5\\}$, and used values for $M_1$ and $M_2$ (from $\Psi_i$) such that the average absolute value of the experiments (across all samples in the time window) would match the simulation, leading to $M_1=4.7748$ and $M_2=14.432$.
>
> **Simulation over the original process**
>
> | Threshold $u$ | Perm. $x_1^*$ | Exc-Inh. $x_1^*$ |
> |---|---|---|
> | 1.5 | $0.0$ | $0.01$ |
> | 2.0 | $0.0$ | $0.01$ |
> | 2.5 | $0.0$ | $0.58$ |
> | 3.0 | $0.0$ | $0.63$ |
> | 3.5  | $0.0$ | $0.63$ |
> | 4.0 | $0.0$ | $0.64$ |
> | 4.5 | $0.0$ | $0.65$ |
>
> **ExtDDM simulation**
>
> | Threshold $u$ | Perm. $x_1^*$ | Exc-Inh. $x_1^*$ |
> |---|---|---|
> | $1.0$ | $0.3$ | $0.0$ |
> | $1.5$ | $0.0$ | $0.0$ |
> | $2.0$ | $0.0$ | $0.7$ |
> | $2.5$ | $0.0$ | $0.7$ |
> | $3.0$ | $0.0$ | $0.7$ |
> | $3.5$ | $0.0$ | $0.7$ |
> | $4.0$ | $0.0$ | $0.7$ |
> | $4.5$ | $0.0$ | $0.7$ |
>
> The results from the last two tables show that, as the threshold grows, we converge to the theoretical optimal nudge time $ x^\*_1 = 0.7 $ for the Excitement and Inhibition scenario (Column 2), and $x^\*_1 = 0$ for the positive drift scenario (Column 1) in both tables.
>
> ## Question 2:
>
> Regarding the train/validation/test splits for the KuaiRec and MOOCCubex datasets, we implemented the following split protocols. For the MOOCCubex we used the split of $70\\%$ for training, $10\\%$ for validation and $20\\%$ for the test data. For the KuaiRec dataset we followed the split protocol used for the public implementation of the EGMN benchmark as presented on Github. It consisted of the $80\%$ split for training and $20\%$ for the test data (note that the Github implementation of EGMN does not use a validation set in its evaluations).
>
> Regarding the other part of your question, you have indeed raised a valid point. However, to properly compute the statistics that you asked us for, we need very extensive computations for the following reasons. To compute the required results for one seed for the KuaiRec and MOOCCubex datasets would take us 6 hours. To produce statistically meaningful results, we need to run our experiments on several seeds (at least 40 to 50 seeds in our estimation). This means that we need at least 240 to 300 hours of computing time; and, unfortunately, we do not have that much time during the rebuttal process.
>
> Therefore, we decided to stick to our current evaluation scheme, which we borrowed exactly from the EGMN implementation in which the seed parameter is set at 42 for the KuaiRec dataset (and we used the same seed in our evaluation experiment).
>
>
> ## Question 3:
>
> Based on your comment, we realized that we did not explain this clearly in the paper. Indeed, the unit diffusion is a rescaling choice: since we estimate the threshold and the drift, for any constant diffusion coefficient $\sigma$ we can obtain the same exit time distribution by using a rescaled process $Z_t/\sigma$ and threshold $u/\sigma$. The diffusion magnitude can be absorbed via rescaling of the drift and threshold parameters. We will revise the paper to make this clear.
>
> Regarding the drift function estimated by the neural network, we calculated descriptive statistics for all samples over the maximum time window:
>
> | Metric | Value |
> |---|---|
> | Mean (all users, all periods) | $1.959411$ |
> | Max (all users, all periods) | $3.338117$ |
> | Min (all users, all periods) | $0.239779$ |
> | Std over time window (per user, avg across users) | $0.010664$ |
> | Std over users (avg over time window) | $0.002586$ |
>
> The drift functions are strictly positive over the entire time window, as the minimum across all users is positive (showing a positive trend towards the decision boundary). We also observe a considerably higher standard deviation over the time window than across samples, which reinforces the empirical gain of the model having a dynamic drift.

---

> > ### Author Rebuttal · Reviewer_N91z · 2026-04-01
> >
> > The rebuttal addresses my concerns, I therefore keep my positive score.

---

> > > ### Author Response · Authors · 2026-04-08
> > >
> > > Thank you for all the constructive and insightful feedback. We are pleased that our response addressed your concerns, and we will certainly incorporate the new results, clarifications, and discussion of rescaling into the final version.

---

### Official Review · Reviewer_2Nvi · 2026-03-08

**Soundness:** 3
**Presentation:** 3
**Significance:** 3
**Originality:** 3
**Overall Recommendation:** 4
**Confidence:** 2

**Summary:**

The paper is motivated by an important gap between cognitively grounded models of decision timing and the demands of modern user-behavior applications. In many platform settings, such as notifications, recommendation feeds, and other interventions, users make decisions under a continuous flow of asynchronous external information. Standard drift-diffusion models provide a natural and interpretable framework for response times, but they quickly become analytically intractable once the drift is allowed to vary over time or depend on external signals. By contrast, point-process and deep learning approaches are often flexible and empirically effective, yet they are typically black-box and offer limited behavioral interpretation. The paper seeks to bridge this gap by developing a tractable model that preserves the cognitive structure of drift-diffusion dynamics while remaining suitable for estimation and intervention analysis.

The model considers a single-agent sequential decision problem over a finite time window. The agent’s latent evidence process evolves as a Brownian motion with a drift term that depends on time-varying external information, and an action occurs whenever this process reaches a threshold, after which it resets. External signals, such as notifications or nudges, enter the model through a kernel-based structure that determines how past signals affect the current drift. The main technical contribution in this paper is a closed-form approximation to the first-passage-time distribution in the high-threshold regime, which gives rise to the proposed Extended Drift-Diffusion Model (ExtDDM). This approximation makes the framework analytically tractable, enables likelihood-based estimation, and allows the authors to study comparative statics and optimization problems concerning the timing of interventions.

**Compliance With Llm Reviewing Policy:**

Affirmed.

**Key Questions For Authors:**

1. Under the Proposition 3.3, the text appears to state that the condition $f(t,u,\xi_t)\to 0$ as $t\to\infty$ can be “easily satisfied” by imposing a positive drift. As written, this seems inconsistent: for a strictly positive constant drift, one would expect $f\left(t, u, \xi_t\right)=\frac{u-\int_0^t \mu(s) d s}{\sqrt{t}} \rightarrow-\infty$ , rather than 0. I may miss something but this point should be checked carefully.
2. It looks to me like the Theorem 3.1 appears to use two different time coordinates in the same integral statement.In the present approximation, the lower integration bound seems to be written in terms of $t$, whereas the upper bound is written as $a-\tau_t$ where $\tau_t$ is the last hitting time before $t$. As written, it is unclear whether the integration variable is meant to run over the original time index or over time measured relative to $\tau_t$. The theorem statement would be much clearer if the authors explicitly fixed one convention and wrote the bounds consistently in that same coordinate system.
3. Is there a typo in Line 240, should it be $h(s)\le 0$?
4. In Theorem 4.2, the probability measure $P_Y$ and $P_Z$ seems not introduced before.
5. In Definition 4.1, the constraint says that $\{x_1, ..., x_J\}\subseteq[0, a]^J$, it seems that you also need $x_1, ..., x_J$ are ordered, right?

**Limitations:**

Please see above

**Strengths And Weaknesses:**

The paper is well motivated and the paper combines drift-diffusion modeling, time-varying external signals, and closed-form first-passage approximation in a nontrivial way that yields a tractable and behaviorally interpretable framework. The provided experiments also align well with the theory. I do not see some significant weakness (but it may be the case that I am not familiar with the related literature).

---

> ### Author Rebuttal · Authors · 2026-03-30
>
> Thank you so much for your careful revision and comments. Our answers to them are provided below.
> ## Question 1:
>
> Thank you for pointing this issue to us. The intended statement "a positive drift will easily satisfy the condition specified in inequality (7)" is true. However, as you correctly indicated, the condition $f\to 0$ clearly is not satisfied for all positive drifts. We realized that we did not explain this point clearly (as in the quoted intended statement above) in the paper and we will revise it to clarify this.
>
> ## Question 2:
>
> Based on your comment, we will revise the paper and will adjust this statement in order to adopt a fixed convention. This should improve readability.
>
> ## Question 3:
>
> As you correctly noticed, there is a typo in the paragraph regarding the definition of $h(s)$ for $s\leq 0$. The correct statement should be $h(s)=0,\forall s\leq 0$. We will revise the paper accordingly.
>
> ## Question 4:
>
> Thank you for pointing this out to us. In the revised paper, we will include the definitions of $P_Z$ and $P_Y$ by defining $Z:=\\{z_i\\}_i^I$ and $Y:=\\{y_i\\}_i^I$, therefore specifying probabilities for each case.
>
> ## Question 5:
>
> In general, the signals do not need to be ordered since they are interchangeable. However, it is possible (and perhaps preferable) to define them as an ordered sequence.

---

> > ### Author Rebuttal · Reviewer_2Nvi · 2026-04-02
> >
> > I would like to keep my score

---

> > > ### Author Response · Authors · 2026-04-08
> > >
> > > Thank you again for your careful review and positive feedback. Your suggestions and requests for clarifications have already helped us improve the clarity and quality of the paper.

---

### Official Review · Reviewer_yacB · 2026-03-13

**Soundness:** 3
**Presentation:** 3
**Significance:** 2
**Originality:** 2
**Overall Recommendation:** 4
**Confidence:** 2

**Summary:**

The paper introduces the Drift-Diffusion Model (DDM), a framework for modeling user decision timing under external signals.
The authors derive a closed-form analytical approximation for the first-passage time distribution of a single-boundary drift-diffusion model with time-dependent drift,  focusing on high-threshold regimes where decisions require prolonged evidence accumulation e.g., purchasing a product after viewing a webpage.  By obtaining a tractable likelihood formulation, the proposed model provides efficient parameter estimation and  external signals, such as notifications or recommendations,  to be incorporated directly into the drift process.
The framework also provides  for optimizing the timing of external interventions (“nudges”) to maximize the probability of user response within a given time window.  Building on this, derive conditions under which certain intervention strategies are more effective. The model is evaluated on two real-world user engagement datasets,  where it shows improved performance over several state-of-the-art methods across multiple evaluation metrics.

**Compliance With Llm Reviewing Policy:**

Affirmed.

**Final Justification:**

I would like to thank the authors for their thoughtful response.
The rebuttal has  addressed my questions and concerns.
In my opinion, overall this paper is good  contribution and is well-suited
for the ICML conference. I therefore support its acceptance.

**Key Questions For Authors:**

1. How accurate is the proposed first-passage approximation for moderate threshold values?
Is it possible to  provide empirical comparisons?

2. The model assumes a single action type (Go/No-Go). Is it possible to extend the framework
 to multi-choice decision processes or multiple event types?

**Limitations:**

Yes

**Strengths And Weaknesses:**

Strengths:

1. The paper is well written and connects with real world examples.

2. The paper derives a closed-form approximation for the first-passage time distribution of a DDM with time-dependent drift in the high-threshold regime.


3. Empirical evaluation on real world datasets.



Weaknesses:


1. The approximation is  valid for the high-threshold case; its accuracy may degrade in low-threshold scenarios.

2. The model assumes a single action type (Go/No-Go).

3. The paper provides the theoretical approximation but does not empirically quantify its error relative to exact simulation-based DDM inference.

---

> ### Author Rebuttal · Authors · 2026-03-30
>
> Thank you for your insightful questions and comments. Our answers are provided below.
>
> ## Question 1:
>
> To provide empirical comparisons of our method depending on the threshold,  we ran an additional experiment over KuaiRec and MOOCCubex demonstrating the connection between the threshold value and MAEs for our method and the EGMN baseline. We produced 244,431 samples and measured the MAE-threshold relationship via (a) linear regression and (b) regression of MAE vs. the log-ratio of EGMN and ExtDDM. Results:
>
> **MAE analysis for KuaiRec**
>
> | Metric | Coeff. - u | p-value |
> |---|---|---|
> | $MAE\\_ExtDDM$ | -0.61 | 0.252 |
> | $\log(MAE\\_EGMN / MAE\\_ExtDDM)$ | 0.93 | 0.000 |
>
> **MAE analysis for MOOCCubex**
>
> | Metric | Coeff. - u | p-value |
> |---|---|---|
> | $MAE\\_ExtDDM$ | 18.93 | 0.070 |
> | $\log(MAE\\_EGMN / MAE\\_ExtDDM)$ | 0.4364 | 0.000 |
>
> $ExtDDM$ MAE decreases with threshold for KuaiRec; the opposite holds for MOOCCubex. However, both p-values (first lines) are insignificant at 5% level, so we performed the comparative analysis (second lines). Those regressions show that EGMN's MAE increases relative to $ExtDDM$'s as the threshold grows, and this is statistically significant.
>
>
> ## Question 2:
>
> Thank you for pointing this valuable extension to us. To answer your question, yes, our method can be extended to modeling multiple choices, as you suggested. However, it can be done under the assumption that each individual "willingness to act" process is mutually independent of the other processes (that are running in parallel). To see this, consider that we have options $\mathcal{S}=\{1,..,J\}$, and we have a random variable $Z\in\mathcal{S}$ indicating the choice made by the agent. Therefore, we can extend our model to this case by obtaining the probability over the chosen option as the choice being the first choice to be taken. More specifically, this would lead to the following probability:
>
> $P(Z=j)=\mathbb{P}\left(\tau_j = \min_{k\in \mathcal{S}} \tau_k \right) = \int_{0}^{\infty} f_j(t) \prod_{\substack{k=1 \\ k \ne j}}^{J} \big(1 - F_k(t)\big) dt.$
>
> where $\tau_j$ is the crossing time, and $f_j$ and $F_j$ are the pdf and cdf, respectively, of the first exit time for option $j$. As the decision threshold of all choices increase, we can use the approximated distribution from ExtDDM for the pdf and cdf, then obtaining the choice probability.
>
> Note, however, that doing this proposed extension properly would require extensive discussion of the independence assumption with respect to the prior literature, which is non-trivial in general. Therefore, we will leave the full study of this extension as a topic for a new standalone paper and will add this observation as a comment in the revised manuscript of our current submission.

---

> > ### Author Rebuttal · Reviewer_yacB · 2026-04-02
> >
> > Thank you for the  response. I would like to keep my score.

---

> > > ### Author Response · Authors · 2026-04-08
> > >
> > > Thank you for your very insightful review and for confirming your concerns are resolved. We will ensure the new analysis and multi-choice discussion are included in the final manuscript.

---

### Official Review · Reviewer_H7oD · 2026-03-13

**Soundness:** 3
**Presentation:** 3
**Significance:** 2
**Originality:** 2
**Overall Recommendation:** 4
**Confidence:** 3

**Summary:**

This paper extends the Go/No-Go Drift-Diffusion Model (DDM) to settings with time-varying drift induced by external signals (notifications, nudges, etc.). The key contribution is a closed-form approximation (Theorem 3.1) for the first-passage time distribution of a single-boundary DDM with time-dependent drift, valid in the high-threshold regime (u -> infinity). This approximation, derived from boundary-crossing results of Cuzick (1981), yields an explicit likelihood that enables scalable inference. Building on this, the authors define the "ExtDDM" process (Definition 3.2) and derive theoretical results for optimal nudge timing (Theorem 4.2, Corollary 4.3) showing that, under high thresholds, optimal signal placement reduces to maximizing cumulative drift near the end of the time window. Empirically, ExtDDM is evaluated on two watch-time prediction tasks (KuaiRec 2.0 short videos and MOOCCubeX educational videos) and on synthetic simulations of four Go/No-Go decision processes, showing improvements over deep learning baselines (EGMN, CREAD, TPM) and neural temporal point process models (RMTPP, NHP).

**Compliance With Llm Reviewing Policy:**

Affirmed.

**Final Justification:**

The rebuttal does not change my evaluation and i will keep my score

**Key Questions For Authors:**

1. **What threshold values does the model learn on KuaiRec and MOOCCubeX, and how does prediction quality degrade as the effective threshold decreases?** The entire theoretical framework rests on u being large. Reporting the distribution of learned u values and showing performance as a function of u (e.g., by stratifying test samples by estimated threshold) would clarify whether the approximation is valid in practice. Without this, it is unclear if the empirical gains come from the DDM structure or from the flexible neural parameterization.

2. **How would the baselines (EGMN, CREAD) perform if given the same 2-step pipeline on KuaiRec?** The hybrid ExtDDM+EGMN approach uses EGMN as a first-stage filter. If EGMN+EGMN (or CREAD+EGMN) with the same 66% quantile split achieves similar gains, then the improvement is attributable to ensembling rather than the DDM formulation.

3. **Can you demonstrate the nudge-timing theory (Section 4) on real or realistic data?** The theoretical results on optimal nudge placement are compelling but entirely disconnected from the experiments. Even a synthetic-but-realistic simulation of notification timing (e.g., using the KuaiRec setting) would bridge this gap and significantly strengthen the paper.

**Limitations:**

yes

**Strengths And Weaknesses:**

**Strengths:**
- The paper bridges cognitive decision theory (DDM) with scalable ML inference in a principled way. The closed-form approximation for the first-passage time density under time-varying drift (Theorem 3.1) is a genuinely useful result — it converts an analytically intractable problem into a tractable likelihood, enabling gradient-based optimization. This is a meaningful contribution to both the DDM and temporal point process literatures.
- The nudge-timing analysis (Section 4) provides clean theoretical insight: under high thresholds, comparing two signal sequences reduces to comparing cumulative drift integrals near the end of the time window (Theorem 4.2). Corollary 4.3 gives concrete prescriptions — send the signal early if its effect is always positive, or time it so its peak hits just before the window ends if it has a refractory period. This connects stochastic control ideas with practical intervention design in a way that is interpretable and actionable.
- The experimental evaluation is thorough and multi-faceted: real-world watch-time prediction on two datasets (Tables 1-2) showing strong improvements (e.g., 75.9% KL improvement on KuaiRec), plus synthetic simulations under four cognitive process variants (Table 3) that validate the high-threshold regime where the approximation excels and honestly reveal where it does not (u=4).

**Weaknesses:**
- The high-threshold assumption (u -> infinity) is central to the entire paper but its practical validity is not rigorously assessed on the real-world datasets. The paper argues that watch-time prediction involves "high thresholds" because decisions take a long time, but this is an informal analogy — there is no diagnostic or calibration showing that the estimated thresholds are actually in the regime where the approximation is accurate. The synthetic experiments (Table 3) show the method loses to IntensityFree in 3 of 4 scenarios at u=4, but the paper does not report what threshold values the real-world model learns or how sensitive performance is to this.
- The experiments validate the ExtDDM likelihood approximation (Theorem 3.1) as a flexible parametric model for event times, but they do not test the paper's two most distinctive claims: the cognitive interpretation (evidence accumulation toward a decision boundary) and the nudge-timing theory (Section 4, Theorem 4.2). Specifically, the general model (Section 3) allows µ(X(t), t) to be any bounded, integrable drift — and the experiments use a valid specialization where the drift depends on static user/video features and smooth time basis functions (polynomials, trigonometric, exponential). This is technically consistent with Theorem 3.1, but it means the "external information" is just static covariates, not dynamic signals arriving during the decision. The kernel structure X(t) = sum h(t - x_j) from Section 4 and its nudge-timing optimization (Theorem 4.2, Corollary 4.3) — arguably the paper's most novel theoretical contribution — are entirely untested. The authors acknowledge this is due to privacy constraints on nudging datasets (Section 5), but it leaves the paper's title and framing ("Under Nudging and External Information") only partially supported by the empirical evaluation.
- The experimental comparisons raise concerns about fairness. On KuaiRec, ExtDDM uses a 2-step hybrid with EGMN (first predicting whether watch time exceeds the 66% quantile, then using ExtDDM only for the "long watchers"). This is a model ensembling strategy that any baseline could also benefit from. The ablation in Appendix B partially addresses this but the main table (Table 1) conflates the benefit of the DDM formulation with the benefit of the 2-step pipeline.

---

> ### Author Rebuttal · Authors · 2026-03-30
>
> Thank you for your insightful questions and comments. Our answers are provided below.
>
> ## Question 1:
>
> We ran an additional experiment over KuaiRec and MOOCCubex demonstrating the connection between the threshold value and MAEs for our method and the EGMN baseline. We produced 244,431 samples and measured the MAE-threshold relationship via (a) linear regression and (b) regression of MAE vs. the log-ratio of EGMN and ExtDDM. Results:
>
> **MAE analysis for KuaiRec**
>
> | Metric | Coeff. - u | p-value |
> |---|---|---|
> | $MAE\\_ExtDDM$ | -0.61 | 0.252 |
> | $\log(MAE\\_EGMN / MAE\\_ExtDDM)$ | 0.93 | 0.000 |
>
> **MAE analysis for MOOCCubex**
>
> | Metric | Coeff. - u | p-value |
> |---|---|---|
> | $MAE\\_ExtDDM$ | 18.93 | 0.070 |
> | $\log(MAE\\_EGMN / MAE\\_ExtDDM)$ | 0.4364 | 0.000 |
>
> $ExtDDM$ MAE decreases with threshold for KuaiRec; the opposite holds for MOOCCubex. However, both p-values (first lines) are insignificant at 5% level, so we performed the comparative analysis (second lines). Those regressions show that EGMN's MAE increases relative to $ExtDDM$'s as the threshold grows, and this is statistically significant.
>
> Threshold distribution statistics (244,431 test samples), with unscaled values on lines 2 and 4:
>
> | Variable | Min | Max | Mean |
> |---|---|---|---|
> | KuaiRec - Normalized | 1.684 | 4.04 | 3.54 |
> | KuaiRec - Seconds | 4.339 | 4.51 | 4.42 |
> | MOOCCubex - Normalized | 1.44 | 3.34 | 2.23 |
> | MOOCCubex - Seconds | 3.12 | 3.31 | 3.29 |
>
> This table shows that the estimated threshold values would be similar to the scenario in the first column of Table 3 of the manuscript, where the performance results are reasonably strong but still weaker than for the higher threshold values reported in columns 2 and 3 of Table 3 of the manuscript. The table above also indicates that rescaling the time variable can lead to changes in the threshold value.
>
> Since we could arbitrarily rescale time, one could question the validity of the approximation (i.e., the rescaling can lead to arbitrary results). Note that this "arbitrariness" conjecture is not true for the following reason. For an increase in the threshold to lead to an improvement in the approximation, we would need to keep other relevant factors, such as the smoothness of the drift, intact (unaffected). And this is not possible with rescaling because this would alter one of the fundamental results from (Cuzick, 1981).
>
> We would like to thank you for your comment because studying the relationship between smoothness of the drift and the threshold value (and other related issues) constitutes an interesting and a highly non-trivial topic that would shed more light on your question of what constitutes a high value threshold. We would like to explore this extensive topic further in a separate paper.
>
> ## Question 2:
>
> We ran the hybrid approach replacing our model with the CREAD model (66% quantile split):
>
> |  | CREAD+EGMN | ExtDDM+EGMN |
> |---|---|---|
> | MAE | 4.443 | 4.283 |
> | XAUC | 0.575 | 0.605 |
> | KL | 0.360 | 0.206 |
>
> The hybrid model underperforms ours on all metrics, confirming our method's viability.
>
> ## Question 3:
>
> We simulated 100,000 Brownian motion paths with nudge-dependent drifts, calibrated to represent KuaiRec's average parameters values. Setup (as in equations (9)-(10) of the manuscript):
>
> - Permanent gain: $h_1(s)=5\exp(-s), s\geq 0$
> - Excitation-inhibition: $h_2(s)=5(0.3-s)\exp(-s), s\geq 0$
> - $\Psi_i(x)=M_i/\pi \arctan(x)$; $u\in\\{1.5,2,...,4.5\\}$; $M_1=4.77$, $M_2=14.43$
>
> **Simulation Original process**
>
> | Thr. $u$ | Perm. $x_1^*$ | Exc-Inh. $x_1^*$ |
> |---|---|---|
> | 1.5 | 0.0 | 0.01|
> | 2.0 | 0.0 | 0.01 |
> | 2.5 | 0.0 | 0.58 |
> | 3.0 | 0.0 | 0.63 |
> | 3.5 | 0.0 | 0.63 |
> | 4.0 | 0.0 | 0.64 |
> | 4.5 | 0.0 | 0.65 |
>
> **ExtDDM simulation**
>
> | Thr. $u$ | Perm. $x_1^*$ | Exc-Inh. $x_1^*$ |
> |---|---|---|
> | 1.0 | 0.3 | 0.0 |
> | 1.5 | 0.0 | 0.0 |
> | 2.0 | 0.0 | 0.7 |
> | 2.5 | 0.0| 0.7 |
> | 3.0 | 0.0 | 0.7 |
> | 3.5 | 0.0 | 0.7 |
> | 4.0 | 0.0 | 0.7 |
> | 4.5 | 0.0 | 0.7 |
>
> As the threshold grows, both setups converge to $x_1^* = 0.7$  (excitation-inhibition) and $x_1^*=0$ (permanent gain), matching theory. We performed this simulation because KuaiRec lacks notification/nudging data. We will clarify this in the revised manuscript.
>
> ## Other Comments:
>
> In the 2nd weakness item, you observed correctly that we mostly used static covariates in the empirical section, and not the dynamic signals arriving at the decision time. First of all, we agree with your observation that having this dynamic signaling capability would improve the empirical analysis of our method. However, unfortunately, we cannot do this properly, not only because of the lack of appropriate datasets on which we could test this approach, but also because benchmarking this method against other models would raise fairness concerns in the following sense. All of our benchmarks do not have a mechanism for changing the distribution during the decision time, which limits the comparison of our method with those benchmarks.

---

> > ### Author Rebuttal · Reviewer_H7oD · 2026-04-03
> >
> > I will keep the score.

---

> > > ### Author Response · Authors · 2026-04-08
> > >
> > > Thank you again for your valuable feedback and for confirming that your concerns have been fully resolved. Your comments have definitely improved the clarity and depth of our work, and we will ensure the new analyses and discussions are reflected in the final text.

---

### Decision · Program_Chairs · 2026-04-30

**Decision:**

Accept (regular)

**Comment:**

All the reviewers agree that the paper has several strong points. First that the paper bridges cognitive decision theory (DDM) with scalable ML inference in a principled way. In particular it is recognized that (Theorem 3.1) is a useful result. At the same time a reviewer appreciate that section 4 provides clean theoretical insight while leveraging the (Theorem 4.2) and Corollary 4.3. The paper is well written and coveys the main message through real world examples and by performing numerical experiments on real world data. Of particular, intereste is the closed-form approximation for the first-passage time distribution of a DDM with time-dependent drift in the high-threshold regime. Numerical experiments are rich and meaningful by including multi-faceted: real-world watch-time prediction on two datasets. However, the reviewers also raised several issues. Numerical experiments only test duration prediction and not if the derived policy actually improves intervention outcomes. Only point estimates are reported, while the KL metric,which is known to be sensitive to binning was not investigated for sensitivity. A theoretical limitation raised by one of the reviewers is relevant, the fact that the approximation is valid for the high-threshold case; its accuracy may degrade in low-threshold scenarios. Furthermore, it seems that the limitation arising from the Go/no go approach represents a concern for some reviewers. Another major issue seems to be the fact that while the manuscript provides the theoretical approximation it does not provide any empiricla quantification under low-threshold scenarios.  An additional criticism is that the manuscript does not clearly provide compute/resource details or a code/artifact statement, which makes reproduction harder. Finally, one reviewer raised a serious issue that  the validity of the  high-threshold assumption is not rigorously assessed on the real-world datasets. Other issues are raised about fairness cowhen coming to numerical experiments.